# The critical role of magma degassing in sulphide melt mobility and metal enrichment

Giada Iacono-Marziano [1 ✉], Margaux Le Vaillant [2], Belinda M. Godel[2], Stephen J. Barnes[2] & Laurent Arbaret[1]

Much of the world's supply of battery metals and platinum group elements (PGE) comes from sulphide ore bodies formed in ancient sub-volcanic magma plumbing systems. Research on magmatic sulphide ore genesis mainly focuses on sulphide melt-silicate melt equilibria. However, over the past few years, increasing evidence of the role of volatiles in magmatic sulphide ore systems has come to light. High temperature-high pressure experiments presented here reveal how the association between sulphide melt and a fluid phase may facilitate the coalescence of sulphide droplets and upgrade the metal content of the sulphide melt. We propose that the occurrence of a fluid phase in the magma can favour both accumulation and metal enrichment of a sulphide melt segregated from this magma, independent of the process producing the fluid phase. Here we show how sulphide-fluid associations preserved in the world-class Noril'sk-Talnakh ore deposits, in Polar Siberia, record the processes demonstrated experimentally.

[1] ISTO, UMR 7327, CNRS, Université d'Orléans, BRGM, F-45071 Orléans, France. [2] CSIRO, Mineral Resources, Kensington, WA 6151, Australia.
✉email: Giada.Iacono@cnrs-orleans.fr

Magmatic sulphide deposits account for significant proportions of the world's production of PGE (>95%) and Ni (>40%)[1] and constitute important sources of Cu and Co. A major 'high quality' deposit of these metals needs to be discovered every year to keep up with the growth in demand from the emerging electric vehicle battery industry. Although a broad consensus exists on the origin of these deposits through segregation, enrichment, and accumulation of immiscible sulphide melts from mafic and ultramafic magmas[2], the mechanisms leading to (1) the accumulation, and (2) the metal enrichment of the sulphide melt are still debated[2–6].

Most magmatic Ni-Cu sulphide ore deposits are characterised by sulphide amounts in large excess over proportions expected for a magma crystallizing along its liquidus surface. This therefore implies efficient mechanisms for sulphide segregation and/or accumulation from the carrier silicate melt[4]. One of the main problems in explaining the formation of sulphide accumulations, such as massive ore deposits, is that the few available quantitative studies[7,8] show that the coalescence of suspended sulphide droplets is not likely in a flowing silicate melt. Analogue experiments illustrate how the coalescence of sulphide droplets is inhibited in flowing magmas (0.01–0.08 m/s), due to the large capillary number (ratio of surface tension force to gravitational body force) of sulphide droplets[7]. Fluid dynamic modeling confirms these results and indicates that droplet break-up, rather than coalescence, is the dominant mechanism for modifying droplet size populations during flow[8]. The accumulation of sulphide droplets may be caused by a reduction in flow velocity, due for instance to a conduit enlargement or a change in flow direction[6]. The sulphide melt may therefore be deposited as a pile of separate droplets, which coalesce gradually after accumulation[7,8], but the operating mechanism is not yet clear[6].

Traditionally, sulphide melts are interpreted as being upgraded —such that their concentration in metals have been increased— by scavenging metals from the silicate melt, as Ni, Cu, Co, PGEs, and Au have a strong tendency to partition into the sulphide melt. The main process considered for the high concentration of these metals in sulphide melt is the interaction of entrained droplets with large masses of silicate melt[1–3,9]. Therefore, one of the major controls on the composition of the sulphide melt is interpreted as being the mass ratio of silicate to sulphide melt, either in a closed system[9], or in an open, replenishing system[2]. Sulfide upgrading is also proposed to be favored by sulphide dissolution, which can be triggered by the interaction with fresh silicate melt that is metal-undepleted and sulphide-undersaturated[10–12]. These processes are however difficult to identify in magmatic rocks because, as pointed out by Mungall and coauthors[11], the occurrence of sulphide upgrading processes cannot be inferred from the composition of the magmatic sulphides.

Hitherto, the role of volatiles in the formation of magmatic sulphide ores has been neglected, due to the generally volatile-poor nature of the parental magmas at the origin of these deposits[13]. However, there is now abundant evidence of the presence of volatiles and their potential importance in mineralizing processes in at least some of these deposits[4,6,14–18]. This presence of volatiles is attributed to either volatile enrichment in magmas at late crystallization stages, or the interaction of the magma with volatile-rich sedimentary wall rocks[4,15–17,19]. Moreover, recent experimental studies show that the sulphide melt is strongly associated with the fluid phase, as soon as it is present in a magma[17,20]; a property well known in metallurgy[21].

In this work, we use both isobaric and decompression experiments at magmatic conditions (i.e., 1150–1200 °C, 50–320 MPa, variable composition of the fluid phase and redox conditions) to constrain the behavior of S during degassing of sulphide-saturated magmas. Our experiments shed light on both physical (coalescence and transport) and chemical (metal enrichment) consequences of the association between sulphide melt and fluid phase.

## Results

**New experiments**. Experiments were performed at pressure-temperature conditions relevant to the emplacement of Nori'lsk-Talnakh magmas, using an olivine gabbro-dolerite from the Noril'sk 1 intrusion[22] as starting composition (Supplementary Table 1). A fluid phase is present in all the experiments, which are divided into two categories, isobaric or decompressed. For each category, multiple experiments were conducted with different starting compositions (different amount and compositions of fluid phase), durations and pressure variations (if any). Details of experimental conditions are presented in Table 1 and the Method section.

All the resulting experimental samples are bubble-bearing and consist of silicate glass, sulphide droplets, and rare olivine crystals (Fig. 1a–d). A rim of PtS formed at the interface with the Pt capsule (Fig. 1a, c). In isobaric samples both olivine crystals and sulphide droplets are concentrated at the capsule side (Fig. 1a), most likely due to Fe-loss in the silicate melt to the Pt capsule, which is however generally lower than 10%.. Sulfide droplets of the decompressed samples are generally larger than those of the isobaric ones (Fig. 1a, c). In decompressed samples, sulphide droplets located in the sample's upper part are larger than those occurring in the lower part of the capsule (Fig. 1c). The sample that experienced the greatest degassing (MLV09) shows sulphide inclusions in olivine crystals, a few corroded sulphide blebs in the upper part of the sample (Fig. 1f), and a Pt-rich, Fe-S-Ni-Cu-bearing phase mainly at the interface between the silicate melt and the olivine crystals (Fig. 1g). The extremely small size of these Pt-rich particles only allowed a qualitative SEM characterisation.

Decompressed samples generally show larger proportions of fluid phase because water exsolves from the silicate melt with decompression, as its solubility primarily depends on pressure[23]. Sulfur degassing is mainly dictated by water exsolution, as its partitioning to the fluid phase increases with the amount of fluid phase[17]. At the explored conditions the fluid phase is therefore expected to be comprised of $H_2O$-$SO_2$-dominated supercritical fluids, with the exception of sample MLV08, in which the fluid phase is Cl-bearing.

The proportion of fluid phase in the experimental samples is observed to be inversely correlated with the proportion of sulphide melt (Fig. 2). Although a significant scatter is observed in the data, both isobaric and decompressed samples seem to share the same trend, with decompressed samples generally showing lower abundance of sulphide melt and higher abundance of fluid phase than the isobaric samples (Fig. 2, Table 1).

We also used an additional sample from a previous experiment[17], in which the fluid phase was generated by the interaction with coal. This sample is significantly more crystallised than the others, with clinopyroxene and plagioclase accompanying olivine (Fig. 1e). In this case, the fluid phase mainly derives from the thermal decomposition of coal, which imposes a $CO$-$H_2S$-dominated composition, and markedly reduces redox conditions, i.e., <FMQ-3, where FMQ is the fayalite-magnetite-quartz oxygen fugacity buffer[17]. The association between sulphide melt and fluid phase is clearly observable in 2D sections of both decompression and magma-coal interaction experiments (Fig. 1c–e).

**High-resolution X-ray computed tomography (HRXCT).** HRXCT analysis was employed to determine the 3D distribution

**Table 1 Experimental conditions of decompression experiments.**

| Sample | H$_2$O$^a$ (wt.%) | Duration (min) | Pressure (MPa) | ΔP/Δt$^b$ (MPa/min) | fO$_2$$^c$ (ΔFMQ) | Fluid phase$^d$ (wt %) | Sulfide melt (wt %) |
|---|---|---|---|---|---|---|---|
| *Without decompression* | | | | | | | |
| MLV01 | 2.0 | 60 | 209 | – | +0.6/+1.6 | 0.13 (7) | 4.8–5.1$^e$ |
| MLV02 | 4.9 | 60 | 209 | – | +1.1/+2.2 | 1.52 (6) | 3.0–3.4$^e$ |
| MLV12 | 4.1 | 180 | 307 | – | +0.2/+0.3 | 0.18 (9) | 3.7–5.8$^e$ |
| MLV15 | 2.0 | 230 | 195 | – | +0.6/+2.2 | 0.10 (7) | 4.6–5.3$^e$ |
| MLV16 | 2.8 | 230 | 195 | – | +1.4/+2.3 | 0.19 (8) | 4.5–4.6$^e$/5.6–6.3$^f$ |
| MLV17 | 3.9 | 230 | 195 | – | +1.7/+2.6 | 0.65 (8) | 3.7–3.9$^e$/4.5–5.1$^f$ |
| *With decompression* | | | | | | | |
| MLV03 | 2.0 | 245 | 202 → 70 | 0.9 | +1.3/+1.6 | 0.72 (8) | 3.6–4.2$^e$ |
| MLV08* | 1.9 | 190 | 204 → 50 | 1.2 | nd | nd | 4.9–5.8$^f$ |
| MLV09 | 4.2 | 330 | 303 → 43 | 2.0 | +1.2/+2.7 | 2.2 (1) | 0.8–0.9$^e$ |
| MLV14 | 4.1 | 262 | 192 → 70 | 1.5 | +1.6/+2.7 | 1.3 (1) | 1.3–2.0$^e$/1.3–1.5$^f$ |
| MLV18 | 2.8 | 262 | 192 → 70 | 1.5 | +2.0/+2.5 | 0.91 (7) | 4.5–4.7$^e$ |
| MLV19 | 3.8 | 262 | 192 → 70 | 1.5 | +1.8/+2.3 | 1.22 (7) | 2.3–2.7$^e$/3.2–3.6$^f$ |

All experiments were run at 1150 °C ± 2 °C.
nd: not determined (quench crystals in the silicate melt may lead to overestimate the fluid phase).
$^a$Amounts of water loaded in the capsule.
$^b$Decompression rates.
$^c$Ranges of oxygen fugacity calculated from V partitioning between olivine and silicate melt using the method of Shiskina et al. 2018 (accounting for the uncertainties in V analyses of both olivine and silicate melt).
$^d$Amount of fluid phase determined by weight loss at the end of the experiment. Numbers in brackets are the errors on the last decimal unit, calculated from the balance uncertainty.
$^e$Amount of sulphide melt determined by mass balance.
$^f$Amount of sulphide melt determined by HRXCT statistics.
*1.7 wt% of NaCl was also loaded in the capsule, no rapid quench.

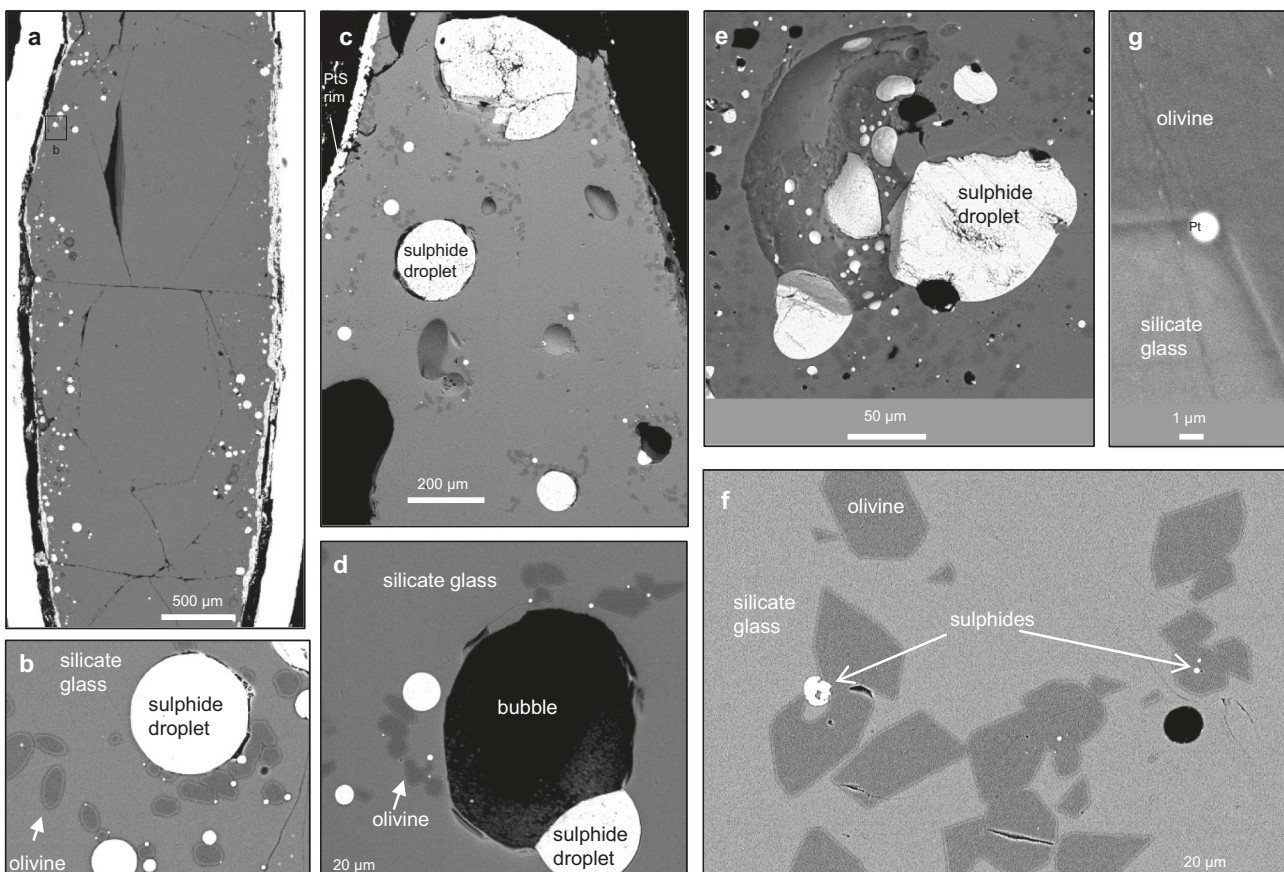

**Fig. 1 Backscatter electron images of experimental samples.** The top of the image represents the top of the capsules. **a** vertical section of sample MLV01 that was equilibrated at 200 MPa and did not experience decompression: it presents low amounts of bubbles and sulphide droplets less than 100 μm in size. **b** Detailed image of sulphide droplets in MLV01. **c** Upper part of sample MLV03 that was decompressed from 200 MPa to 50 MPa, containing sulphide accumulations of several hundreds of microns. **d** Detailed image of a compound drop in MLV03. **e** Detailed image of a compound drop in sample #6.3. **f** Sample MLV09 that was decompressed from 320 MPa to 50 MPa. **g** Detailed image of a Pt-rich phase (Pt) at the interface between the olivine and the silicate melt in MLV09.

of bubbles and sulphide globules inside the silicate glass and the volume proportions of silicate glass (+olivine), sulphide globules and bubbles. HRXCT reveals that the sulphide melt is generally associated with the fluid phase, independently of its composition. Most of the sulphide droplets are connected to fluid bubbles, except those that are in contact with the PtS rim at the capsule interface. Figure 3 shows the 3D distribution of sulphide droplets and fluid bubbles in the most representative decompressed sample (a larger proportion of the sample was imaged; Fig. 3a–c), and the sample that experienced interaction with coal (Fig. 3d–g).

HRXCT renderings show that the sulphide droplets connected to the same fluid bubble may coalesce, forming larger blebs (Fig. 3c). The size of the sulphide blebs is generally larger in decompressed than in isobaric samples (Fig. 1a, c). HRXCT data show that the sulphide blebs within decompressed samples attain larger volumes (Fig. 4a) and are less numerous (Fig. 4b) than those in isobaric

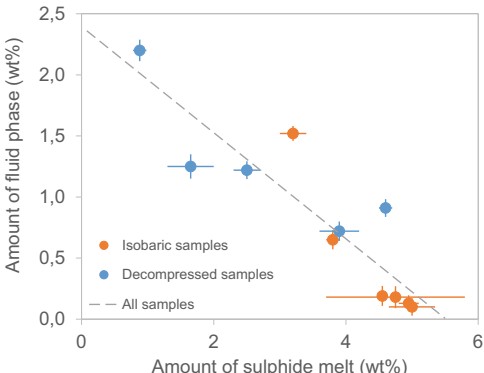

**Fig. 2 Weight proportions of sulphide melt and fluid phase in experimental samples.** Error bars represent the standard deviation around the mean values. The same inverse correlation is observed in both decompressed (blue points) and isobaric samples (orange points).

samples (see also Supplementary Table 4). Moreover, decompressed samples show the occurrence of large bubbles and sulphide blebs in their upper part (Figs. 1c and 3a). In the sample that interacted with coal the largest sulphide blebs are randomly distributed and have irregular tortuous shapes (Fig. 3d, e), with bubbles entrapped inside (Fig. 3e–g).

**Phase compositions.** Compositions of the silicate glasses, sulphide droplets and olivine crystals occurring in the experimental samples are available in Supplementary Tables 1–3. Figure 5 illustrates that the average Ni and Cu contents of the sulphide melt increases with decreasing sulphide content. Decompressed samples generally show higher Ni and Cu contents and a larger variability than isobaric ones (Fig. 5).

**Discussion**

The strong association between sulphide melt and the fluid phase shown by HRXCT analysis (e.g., Fig. 3) confirms previous observations that when a fluid phase is present in a magma, most of the sulphide droplets are connected to fluid bubbles, forming compound drops[20]. This is the result of the tendency for heterogeneous nucleation of sulphides and bubbles[11,24]. In our samples sulphide droplets that are connected to the same fluid bubble tend to coalesce, commonly while sliding toward the lower part of the bubble (Fig. 3c). This increases the size of the sulphide droplets and decreases their number, as clearly observed when comparing isobaric and decompressed samples (Fig. 4). This coalescence of sulphide droplets to form larger sulphide blebs is a crucial, previously unnoticed implication of compound drops. Coalescence of sulphide droplets may be facilitated by the lowering of their interfacial tension induced by the bubble. However, the main driver for coalescence to occur is likely to be the fact that connection to the bubbles keeps the droplets in contact for a long enough time to allow drainage of the melt film between them, as opposed to the situation in a flowing magma where adjacent droplets are sheared apart before the melt film has time

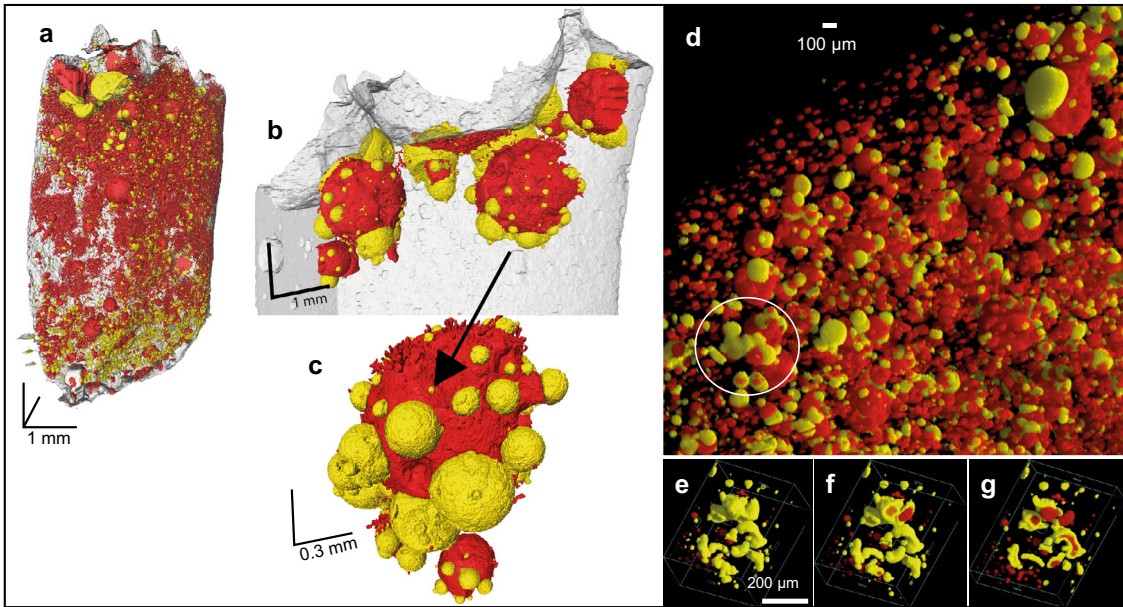

**Fig. 3 HRXCT renderings of representative experimental samples.** 3D distribution of sulphide droplets (in yellow) and fluid bubbles (in red) are shown for samples MLV08 (**a–c**) and #6.3 (**d–g**). **a** Half of the sample, without the Pt capsule. The silicate melt is in gray and the top of the image indicates the top of the capsule. **b** Detailed image of the upper part of the sample. **c** Detailed image of compound drops showing several sulphide droplets attached to the same fluid bubble. **d** Sample #6.3 that interacted with coal. The shape of the sulphide bleb in the circle strongly indicates coalescence. **e** Shape of the largest sulphide bleb. The fluid phase is shown in red with the same segmented volumes of the sulphide bleb. **f, g** The segmented volumes of sulphide blebs are smaller than the one of fluid bubbles, in order to hide a part of the sulphide blebs and to show the bubbles occurring at their interior.

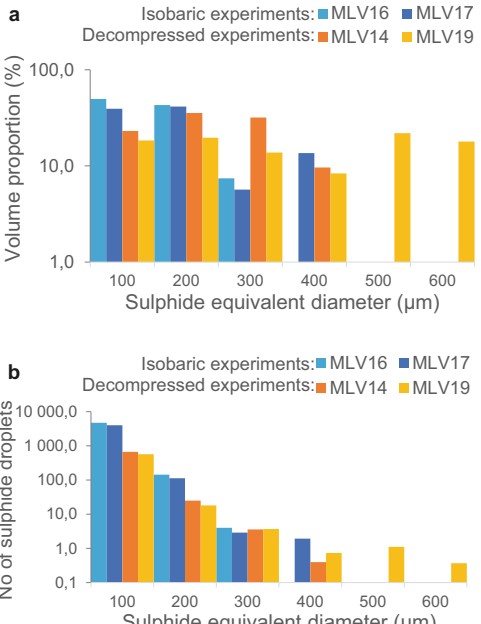

**Fig. 4 HRXCT data. a** Distribution of the volumes of the sulphide blebs in two isobaric (MLV16 and MLV17) and two decompressed samples (MLV14 and MLV19), and **b** number of sulphide droplets (per volume unit) for each size of sulphide blebs (in equivalent sphere diameter).

to drain[8]. This process has been investigated for coalescence of gas bubbles in lavas, a good analogy because of the similarity in capillary number between sulphide droplets and vapor bubbles[25]. The larger the sulphide bleb, the larger its capillary number, and the greater its ability to deform and facilitate coalescence. The composition of the fluid phase and the redox conditions may control the morphology of the interface between sulphide droplets and bubbles[20,26], but they do not seem to deeply affect droplet coalescence, as sulphide accumulations are systematically observed in all decompressed samples and in the magma-coal interaction experiment (Figs. 1 and 3).

As proposed by the previous studies[6,20,26], sulphide droplets can be transported upward in a magma by fluid bubbles, if the bulk density of the compound drop is lower than that of the silicate melt. The degree of crystallization of the magma is likely to be a crucial parameter controlling the mobility of the compound drops in the magma[6,26]. We indeed observe the occurrence of larger bubbles and sulphide blebs in the upper part of decompressed samples (Figs. 1c and 3a), indicating that compound drops can coalesce during their upward transport in these crystal-poor silicate melts, increasing both the bubble and the droplet size. When the crystal content of the magma is higher, the upward movement of the sulphide melt is impeded, and both bubbles and sulphide blebs can be deformed. This is illustrated by our experimental sample that is significantly more crystallised due to interaction with coal[17] (Fig. 1e), which presents large sulphide blebs with irregular shapes (Fig. 3d, e). This suggests that crystallization may limit the mobility of compound drops, but also facilitate their contact and therefore promoting sulphide coalescence. In a crystal-rich magma, the upward transport of sulphide droplets would therefore be limited and their accumulation favored.

In addition to these crucial implications for sulphide melt accumulation, the formation of compound drops during magma degassing may also have consequences for the metal enrichment of the sulphide melt. Mungall and coauthors[20] proposed that the

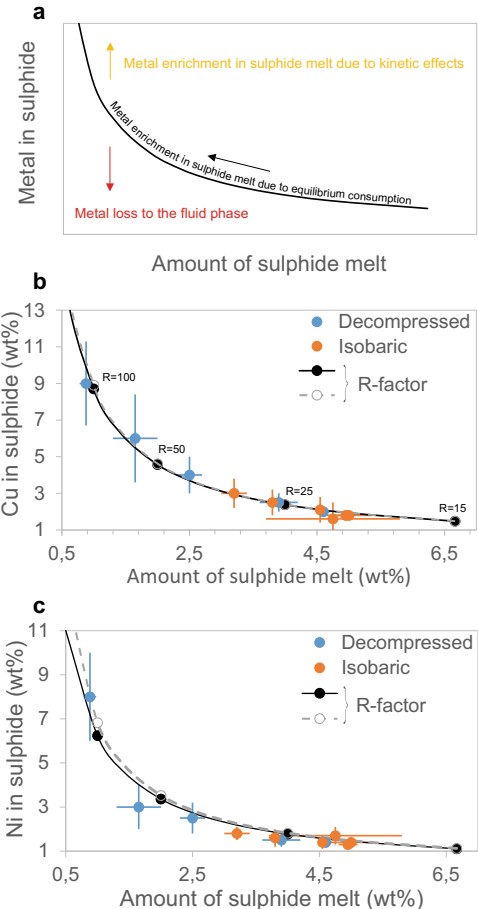

**Fig. 5 Metal content of the sulphide melt. a** Theoretical scheme showing how sulphide consumption at equilibrium conditions increases the metal content of the sulphide melt, following the black line, which is calculated after ref. [9]: $C_{sulph}=C° D (R+1)/(R+D)$, where $C_{sulph}$ is the metal concentration in the sulphide melt, $C°$ is the metal content of the silicate melt before sulphide segregation, $D$ is the metal partition coefficient between sulphide and silicate melts, and $R$ is the silicate/sulphide mass ratio. Departure from equilibrium conditions could be due to either metal transfer to the fluid phase (red arrow), or metal enrichment in the sulphide melt due to delayed diffusion of metals (green arrow). Average Cu (**b**) and Ni (**c**) contents of the sulphide melt versus its amount in each sample determined by mass balance calculations. Error bars represent the standard deviation around the mean values. The solid and the dashed lines indicate equilibrium contents for variable silicate/sulphide mass ratios (indicated on the line), as in (**a**), calculated for the minimum (solid line) and the maximum (dashed line) $D_{sulf/sil}$ values determined in this study.

rise of compound drops in magmas leads to sulphide melt consumption, due to S extraction to the fluid phase. The inverse correlation between the amounts of sulphide melt and fluid phase (Fig. 2) clearly points to sulphide melt consumption due to S degassing to the fluid phase, in agreement with thermodynamic constraints indicating that S partitioning to the fluid phase increases with fluid-phase abundance[17]. This confirms the strong affinity of S for the fluid phase[27] and suggests that the occurrence of a fluid phase decreases sulphide melt stability, independent of the occurrence of decompression (decompressed and isobaric samples share the same trend). The scatter observed in the experimental data in Fig. 2 is likely explained by the variable redox conditions used in the experiments (Table 1), which impact S partitioning between the silicate melt, the fluid phase, and the sulphide melt. Pressure may also play a role in S degassing by

affecting the S content at sulphide saturation: with the decrease of pressure, the S content at sulphide saturation increases[28], and S may be partially redissolved into the silicate melt, contributing to sulphide consumption. However, we do not observe any increase in the S content of the silicate melt of decompressed samples, indicating that degassing prevails over redissolution at the investigated conditions. The S content of the silicate melt varies between 0.03 and 0.6 wt% in our samples (Supplementary Table 1) and is not correlated with the amount of sulphide melt.

Concurrent with sulphide consumption our experimental results clearly show metal enrichment in the sulphide melt (Fig. 5). Sulfide dissolution due to interaction with sulphide-undersaturated magma (i.e., during magma recharge) has already been proposed to upgrade the metal content of the sulphide melt[10,11]. We propose that sulphide consumption driven by magma degassing also triggers sulphide upgrading. The upgrade is less important in this case because no metals are added from the silicate melt; they are only concentrated into the sulphide melt by S loss. Metal enrichment may nonetheless attain high tenors when sulphide consumption is pronounced (Fig. 5). For our isobaric samples we calculate sulphide melt/silicate melt partition coefficients of 684-2125, 1231-1857, and 58-142 for Ni, Cu, and Co, respectively, which are in the range of previous literature values[29–31]. Using these partition coefficients, we compute the Ni, Cu, and Co content of the sulphide melt, as a function of the mass ratio (i.e., R-factor) of silicate to sulphide melt (solid lines in Fig. 5), following the equation of Campbell and Naldrett[9]. Experimental data show that a diminishing mass of sulphide melt equilibrates with a constant mass of silicate melt, increasing R and hence increasing metal content (Fig. 5, Supplementary Table 2). Decompressed samples attain the highest proportions of fluid phase and therefore the highest Ni and Cu contents in the sulphide melt (Fig. 5 and Supplementary Fig. 1). The larger variability observed in these samples suggests the occurrence of disequilibrium processes during decompression and degassing. Figure 5a illustrates that the two disequilibrium processes expected to operate in these conditions are (i) the metal loss to the fluid phase, and (ii) the kinetic enrichment of the sulphide melt. The former would transfer part of Ni and Cu to the fluid phase[16,20] and therefore reduce the metal content of the sulphide melt, whereas the latter would increase the metal content of the sulphide melt, due to a delayed diffusion of metals back to the silicate melt. The large data variability does not allow us to determine which of the two processes is the more dominant.

In the experimental sample (MLV09) that experienced the greatest degassing (i.e., highest proportion of fluid phase, >2 wt %), the consumption of the sulphide melt is the most advanced and sulphides are mainly observed as inclusions in olivine crystals (Fig. 1f). This agrees with the occurrence of sulphides only as inclusions in the phenocrysts and not in the groundmass, in natural volcanic samples that have undergone extensive degassing[32,33]. The few corroded sulphide blebs observed in the upper part of sample MLV09 (Fig. 1f) show metal contents up to 8 ± 2 wt% Ni and 9 ± 2 wt% Cu (Supplementary Table 2). These contents are higher than those of the other samples, and indicate an R-factor larger than 100 (Fig. 5b, c). A Pt-rich phase is also observed in the same sample (Fig. 1g), consistent with previous experimental observations indicating that sulphide degassing can stabilize magmatic Pt minerals[34]. These Fe, S, Ni, and Cu-bearing Pt micro-nuggets could represent an artefact due to Pt excess imposed by the Pt capsule. The fact that they are observed uniquely in this sample (whereas all the samples have Pt capsules) suggests that extreme degassing of the sulphide melt may lead to the formation of PGM. The relatively low Ni, Cu, and Co content of the silicate melt of this sample (Supplementary Table 1) suggests that, when the sulphide melt is exhausted, metals that are

not concentrated in PGM could be transferred to the fluid phase rather than redissolved into the silicate melt, as proposed by Mungall et al.[20].

An origin of PGM by partial desulphurisation of Ni-Cu sulphides at relatively low temperature has been invoked for the Merensky Reef of the Bushveld Complex (South Africa)[35]. If occurring at higher temperatures, similarly to our experiments, this process may account for the occurrence of noble metal-bearing nanoparticles associated with sulphide and fluid inclusions within olivine phenocrysts of recent basaltic products of the Tolbachik volcano (Kamchatka arc, Russia)[36], and the PGE-rich ores described in the next section.

Transport, coalescence and metal enrichment of sulphide droplets are three key components necessary for the generation of a magmatic Ni-Cu-Co-PGE ore deposit, and all three of those seem to be potentially affected by the presence of compound drops. Our results particularly highlight how the implications of compound drops depend on the proportion of fluid phase (Fig. 6a), regardless of whether it is due to magma degassing or to crustal assimilation. Low proportions of fluid phase favor the coalescence of the sulphide droplets and therefore the accumulation of the sulphide melt; when the amount of fluid phase is higher, the consumption of the sulphide melt dominates, accompanied by metal enrichment; extremely high degassing may trigger the formation of PGM. Coalescence of sulphide droplets, sulphide melt consumption and PGM formation may occur in different parts of the same magmatic body, leading to the formation of distinct types of magmatic sulphide ores, as illustrated by one of the most valuable metal concentrations on Earth: the Noril'sk-Talnakh ores in polar Siberia. These ores are hosted in mafic-ultramafic subvolcanic ribbon-shaped intrusions[2,22]. Experimental results presented here were obtained using starting melt compositions similar to the parental melts of the Noril'sk-Talnakh ore hosting intrusions and were conducted at matching pressure and temperature conditions. Textures and ore compositions observed within the different ore types in Noril'sk-Talnakh intrusions can be explained through the processes that have been described above (Fig. 6). Noril'sk-Talnakh three main ore types are: (i) low-sulphide PGE ores in the upper part of the intrusion within the Upper Gabbroic Series[22,37–39], (ii) disseminated sulphides, or globular ores inside picritic and taxitic rocks, in the lower part of the intrusion[2,15,16,40,41], and (iii) massive sulphide ores also in the lower part of the intrusion, largely in the country rocks and less frequently in the intrusion itself, occasionally accompanied by Cu-rich "cuprous ores"[2,22,42] (Fig. 6b, c). Textures and compositions observed in these three ore types can be linked with the interaction between the sulphide liquid and the fluid phase. In the first two types of ores subspherical structures within the crystalline framework (Fig. 6b) have been interpreted as fluid bubbles filled with late magmatic phases or hydrothermal minerals[15,16,37–39]. In the lower part of the intrusion, these structures are systematically associated with sulphide minerals suggesting they represent compound drops that were frozen in place in the olivine-rich rocks[15,16,37]. In the upper part of the intrusion, these subspherical structures are even more common and generally contain lower amounts of sulphide minerals but abundant PGMs and a very high PGE/S ratio[38,39,43]. In this ore type, the PGE-rich minerals are displaced from the residual sulphide droplets, which is consistent with sulphide dissolution[12]. Finally, the massive sulphides are interpreted as internally differentiated accumulation of sulphide melt[2,42]. We propose that the massive accumulation of sulphide melt in Noril'sk-Talnakh intrusions was facilitated by its association with the fluid phase. The compound drops that accumulated to form the massive sulphides are those that experienced limited flotation and rapidly settled, due to their bulk density being higher than that of the

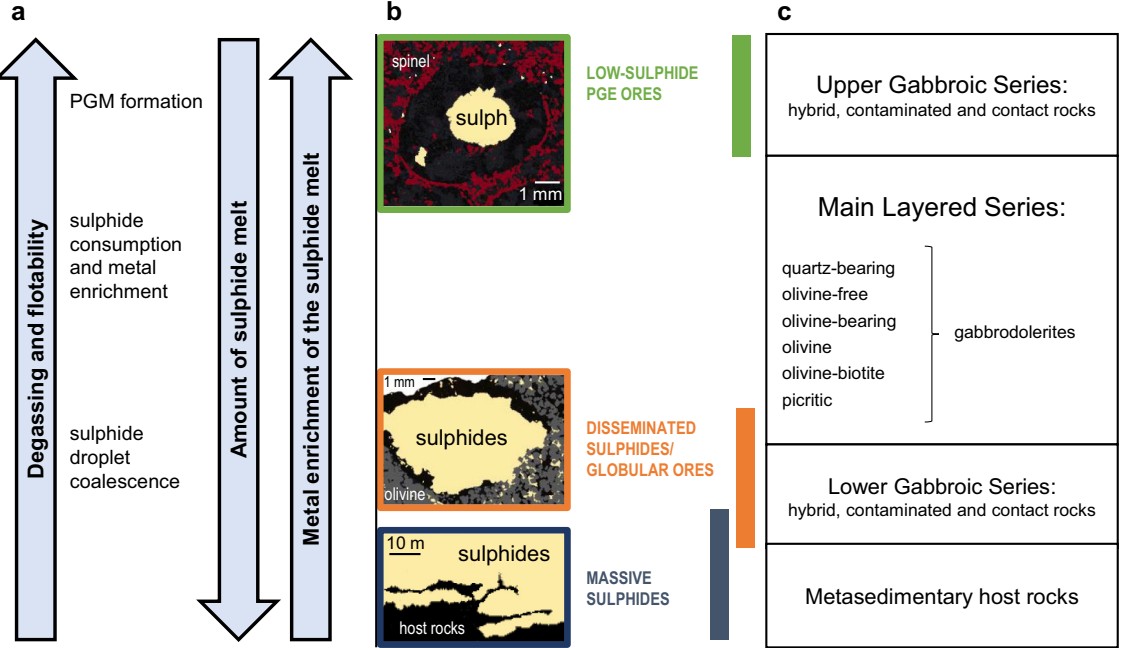

**Fig. 6 Proposed scenario for ore type formation in Noril'sk-Talnakh intrusions. a** Scheme showing how floatability and metal enrichment of the sulphide melt increase with increasing degassing, whereas the amount of sulphide melt decreases. **b** Magmatic sulphide ore types in Noril'sk-Talnakh deposits and (**c**) their distribution within the intrusion. From top to bottom in (**b**), schematic drawings of: the subspherical shapes described in low-sulphide PGE ores within the Upper Gabbroic Series (as observed in refs. [39,40]); the subspherical shapes in globular ores within the picritic rocks of the Main Layered Series and the taxitic rocks of the Lower Gabbroic Series (as observed in refs. [15,16]); massive sulphide ores within the metasedimentary host rocks (as observed in refs. [22,41]).

silicate melt, or due to physical detachment of large droplets from bubbles[6]. No remnants of bubbles are found in these ores, and the fluid phase likely escaped into the host rocks contributing to the formation of the metasomatic aureoles, largely described in Noril'sk-Talnakh intrusions[2,40,44,45]. The lower extent of degassing experienced by massive sulphides is attested by the lower metal contents with respect to disseminated sulphides[2,42]. Figure 7 shows Ni and Cu contents of massive and globular ores (recalculated to 100% sulphide) in Noril'sk-Talnakh intrusions[2,42]; data for low-sulphide PGE ores are not available. Globular sulphides (i.e., disseminated ores) show larger variations than massive ones, but the average values indicate higher Ni and Cu contents in globular sulphides. We consider the hypothesis that variability between ore types in the Norilsk-Talnakh deposits could be a result of decreasing sulphide liquid mass due to S loss during degassing, corresponding to an increase in silicate/sulphide mass ratio (i.e., R-factor). A direct comparison with experimental data would not help interpreting the observed variations, because Ni and Cu contents of our starting product (Supplementary Table 1) are likely to be higher than those of Noril'sk parental magmas. We therefore calculated how Ni and Cu contents of the sulphide melt vary when the amount of sulphide melt decreases (i.e., R-factor increases), using the equation of Campbell and Naldrett[9]. We considered a parent magma with 120 ppm Ni and 150 ppm Cu, which are the maximum contents measured in the basaltic lavas of the Noril'sk region, and are likely to represent Ni and Cu contents of the parental magma of ore-bearing intrusions[2]. We used Ni and Cu sulphide-silicate partition coefficients that we determined for the most reducing and the most oxidising redox conditions (Table 1), in order to account for the likely variability in Noril'sk-Talnakh intrusions[17]. Calculated trends suggest that globular ores formed under more oxidising conditions than massive ores (Fig. 7). Average Ni and Cu contents of both massive and globular sulphides fall close to the calculated trends and indicate R-factors of 400 and 1300,

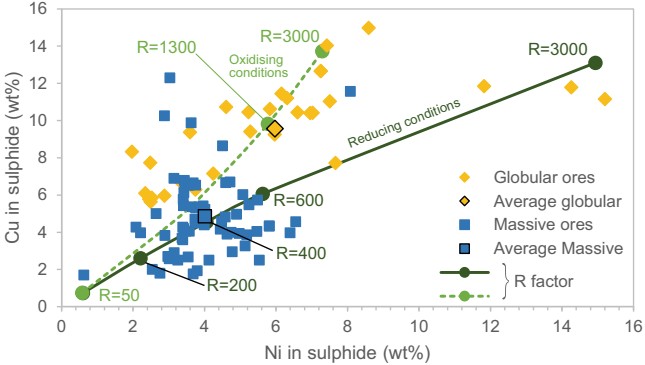

**Fig. 7 Ni and Cu contents of globular and massive sulphide ores in Noril'sk-Talnakh intrusions.** Data for globular sulphides (in yellow) are recalculated to 100% sulphide and are from ref. [2]; the average of 30 analyses is shown. Data for massive sulphides (in blue) are from refs. [2,43]; the average of 64 analyses is shown. The solid and the dotted lines indicate equilibrium contents for variable silicate/sulphide mass ratios, i.e., *R* (indicated on the lines), calculated following Campbell and Naldrett's equation[9] using Dsulf/sil values determined in this study for the most reducing and the most oxidising redox conditions (Table 1), and initial Ni and Cu concentration in the parental magma of 120 ppm and 150 ppm, respectively.

respectively (Fig. 7). R-factors up to 3000 account for the highest Ni and Cu contents observed in disseminated sulphides (Fig. 7). If we assume that S degassing is the only operating mechanism (without any interaction of the sulphide melt with metal-undepleted silicate melt), these values of R-factor would suggest that globular sulphides experienced 69% of consumption on average by S degassing, and up to 87% to account for the highest Ni and Cu tenors. These extreme values would therefore

represent the maximum Ni and Cu contents attained in Noril'sk-Talnakh ores, before the total consumption of the sulphide melt and the crystallization of PGM. However, we stress that S degassing is unlikely to occur in the absence of sulphide melt-silicate melt interactions, because compound drops favor the transport of sulphide droplets within the silicate melt, contributing to increase the R-factor. In this scenario, lower degrees of sulphide consumption are necessary to explain metal tenors of Noril'sk-Talnakh ores.

We did not analyse PGE in our samples, due to the presence of the capsule that is an additional Pt reservoir preventing mass balance calculations. However, we performed similar calculations for Pt and Pd using sulphide-silicate partition coefficients from the literature[30] (see Supplementary Information). Results obtained for Pt and Pd are in a good agreement with those obtained for Ni and Cu (Supplementary Fig. 2), and corroborate R-factors estimated for massive and globular ores using Ni and Cu. Estimated PGE tenors for the sparsely disseminated sulphides associated with the vesicular upper taxites, the so-called low-S high PGE ores, are in the range of 200–2000 ppm Pt and Pd based data presented by ref. [39]. Textural data strongly suggests that these ores underwent extensive degassing accompanied by loss of S and formation of PGMs[38]. These extreme tenors could strongly suggest PGM formation by sulphide loss due to degassing, although additional processes such as redissolution of sulphide into undersaturated silicate melt cannot be excluded.

We conclude that ore distribution in Noril'sk-Talnakh intrusions are indicative of an increasing extent of degassing from the bottom toward the top of the intrusions, implying increasing sulphide melt consumption and metal enrichment (Fig. 6). The occurrence of an abundant fluid phase during ore formation is described in these ore deposits (see ref. [17] for a review) and attributed to the interaction of the magma with evaporitic and carbonaceous rocks[2,17,46,47]. The main markers of fluid involvement are intrusive breccias and pegmatoidal/taxitic units[4,39,48], and intense metamorphic/metasomatic aureoles[2,40,44,45]. Although a role for volatiles is less clear in other magmatic sulphide deposits, an increasing number of examples of sulphide-fluid associations are reported[14,18,49]. Similar to those observed in Noril'sk-Talnakh intrusions, spherical textures associated with silicate-filled cavities are described in two komatiite-hosted magmatic sulphide ores, the Black Swan disseminated ores (Western Australia) and the Alexo deposit (Ontario); they are interpreted as amygdales or segregated vesicles[49]. Spherical sulphide globules displaying a coarse-grained silicate cap have been reported (i) in other komatiitic flows of Western Australia[50] and Canada[51], (ii) in intrusions associated with the Karoo flood basalts in South Africa[52], and (iii) in mafic dykes in East Greenland[53] and Uruguay[54], and also interpreted as segregation vesicles[49]. We therefore suggest that the mechanisms we describe in our experiments may be a lot more common than currently considered.

Different processes cause volatile saturation and exsolution in magmas, the most common ones being decompression, crystallization and interaction with volatile-rich sedimentary rocks. Decompression during magma ascent triggers the exsolution of volatile species, like $H_2O$ and $CO_2$, whose solubility depends heavily on pressure[23]. As volatile species are generally incompatible with the major crystalline phases, they are concentrated in the silicate melt by crystallization and may reach saturation levels. This process is responsible for fluid phase exsolution in crystal-rich magma[55]. In addition, assimilation of volatile-rich sedimentary rocks may contribute external volatiles and trigger the formation of a fluid phase even in volatile-poor primitive magmas. The Noril'sk-Talnakh intrusions are probably the best documented case[15–17,37–39], but several other magmatic sulphide ores present evidence of the occurrence of a fluid phase during ore formation, independently of the crustal or mantle origin of the volatiles. Evidence of volatile-rich environment has been, for instance, postulated in the three layered intrusions hosting PGE-rich magmatic sulphide layers, the Bushveld Complex (South Africa)[56–59], the Stillwater Complex (USA)[56,58] and the Lac des Iles Complex (Canada)[2,60], although none of the processes described here has been ever invoked for their formation.

We conclude that the role of compound drops in both facilitating sulphide accumulation and attaining high metal enrichments should probably be carefully evaluated for several magmatic sulphide deposits other than Noril'sk-Talnakh. Particularly the interaction with S-bearing crustal rocks, in addition to be instrumental for sulphide melt segregation[2], may allow the release of large amounts of fluid phase, and therefore favor the processes we describe in our experiments.

Moreover, compound drops are proposed to promote sulphide mobility also in the lower crust: the association of mantle-derived carbonates and sulphides has been observed in some mafic-ultramafic magmatic systems and attributed to the formation of compound drops involving supercritical $CO_2$ fluids[18]. Our results focus on upper crust, $H_2O$-dominated conditions and suggest that the amount of fluid phase is the first order parameter in determining S degassing and therefore sulphide consumption. If this is confirmed for $CO_2$-dominated systems and higher pressures, $CO_2$ degassing is likely to have a weaker effect on S degassing, as $CO_2$ solubility, even at lower crustal conditions, is generally lower than $H_2O$ solubility[23]. However, the composition of the fluid phase may also play an important role and devoted experiments would be required to clarify it, as existing degassing models do not account for the sulphide melt.

Furthermore, we suggest that the process of enhanced coalescence of fine droplets attached to bubbles may be a more general solution to the unsolved problem of how sulphide droplets are deposited from flowing magmas to form Ni-Cu-Co dominant deposits. Transport of large droplets in rapidly flowing magmas should lead to droplet breakup and entrainment as a fine emulsion[8], but if droplets are "harvested" by bubbles they could coalesce, detach and settle. Such a process could very well be associated with transient degassing at dyke-sill transitions[61], one of the most common loci for sulphide deposition[4].

## Methods

**Experimental methods.** An olivine gabbro-dolerite from Noril'sk 1 intrusion (drill hole OM-1, depth 1005 m[22]) was powdered and melted at 1600 °C and atmospheric pressure for 2 h, in order to obtain a homogeneous, volatile-free glass. This glass was used as starting material for the experiments.

Decompression experiments were designed to simulate volatile release from a magma in response to a pressure decrease. They cover variable extents of volatile degassing and variable redox conditions (Table 1). In these experiments starting glass powder was loaded into Pt capsules (internal diameter 2.5 mm) together with 5 wt% elemental sulfur and 1.9-4.9 wt% $H_2O$ (Table 1). In one sample 1 wt% chlorine was also added in the form of NaCl. All experiments were conducted at 1150 °C, in internally heated pressure vessels equipped with rapid quenching devices. Six samples were equilibrated at isobaric conditions (between 195 and 307 MPa), whereas six other samples were firstly equilibrated at isobaric conditions (between 192 and 303 MPa), and then decompressed to 43-70 MPa, with decompression rates between 0.9 and 2.0 MPa/minute (Table 1). Redox conditions were varied by adjusting the partial pressure of hydrogen in the vessel, and calculated from V partitioning between olivine and silicate melt using the methods of Shiskina et al.[62] (Table 1). All samples were rapidly quenched with a rate of ~100 °C/s, except for sample MLV08, for which the quench rate was slower. Sample MLV08 was not analysed for phase compositions, and this allowed a larger portion of the sample to be characterised by computed tomography to illustrate sulphide melt-fluid phase relationships.

Magma-coal interaction experiments are presented in ref. [17]. One sample is here investigated in details, #6.3: it was obtained by loading a $H_2O$-bearing, sulfate-saturated glass (2.1 ± 0.1 wt% $H_2O$ and 0.9 ± 0.1 wt% S) into a Pt capsule, together with chips of coal from the Noril'sk region, Russia[17]. The experiment was conducted at 1200 ± 2 °C, and 75 ± 7 MPa and lasted 145 min.

**Analytical methods.** The amount of fluid phase occurring in each sample was determined by weight loss upon piercing of the experimental capsule at the end of the experiment. Major element compositions of experimental glasses, minerals, and sulphide droplets were analysed with a Cameca SX Five electron microprobe at the

Institut des Sciences de la Terre d'Orléans (Orléans, France). The following operating conditions were used for glasses and silicate minerals: 15 kV accelerating voltage, 6 nA beam current, 10 s counting time for all elements on each spot (except for S: 120 s), 10 μm spot size for glasses and a focused beam for minerals. Sodium and potassium were analysed first to limit any losses. Sulfide droplets were analysed using 20 kV, 30 nA, 10 s counting time for Si, S, Fe, Ni, Cu and 120 s for O. The beam sizes varied depending on the size of the sulphide.

Ni, Cu, and Co concentrations in the silicate glasses were determined by laser ablation inductively coupled plasma mass spectrometry (LA-ICP-MS) at the Institut des Sciences de la Terre d'Orléans (Orléans, France), using an Agilent 8900 QQQ mass spectrometer coupled to a RESolution Excimer laser (ArF 193 nm), at the IRAMAT-CEB in Orleans. The following masses were measured: $^{29}Si$, $^{44}Ca$, $^{43}Ca$, $^{47}Ti$, $^{57}Fe$, $^{59}Co$, $^{60}Ni$, $^{61}Ni$, $^{63}Cu$, $^{65}Cu$. Co, Ni, Cu concentrations were quantified using the NIST SRM 610 glass[63]. $^{44}Ca$ was used as internal standard, a laser beam of 100 μm diameter and a frequency of 10 Hz were employed. Two reference materials were used for quality control: the NIST SRM 612 synthetic glass[63] and the BCR-2G USGS natural glass[64]. Data reduction was performed using GLITTER software.

**High-resolution X-ray computed tomography (HRXCT).** HRXCT analyses were performed at the Australian Resources Research Centre (Kensington, Western Australia) using a Zeiss Versa XRM 520 3D X-ray microscope. Mosaic scans were acquired (along the vertical axis) to image the entire capsules using 1601 projections over 360° rotation with accelerating voltage of 160 kV, a power of 10 W, a current of 62 μA and a voxel size of 3 μm. After reconstruction, each volume was quantified using methods described in ref. [65]. The weight proportions of the sulphide globules (Table 1) were calculated from the volumes of the different phases, using densities ranging from 2.52 to 2.75 g/cm³ for the silicate glass (calculated from the silicate glass composition at the glass transition temperature following ref. [66], 4.0 to 4.5 g/cm³ for the sulphide globules[67]. A density of the gas bubble varying between 0.1 and 1 g/cm³ only affects the estimation of the sulphide proportions negligibly, whereas the uncertainty on the sulphide density is the main controlling factor. The HRXCT analysis of Sample #6.3 was performed at the Institut des Sciences de la Terre d'Orléans (Orléans, France) using a Nanotom 180NF Phoenix X-Ray. The machine operates with an accelerating voltage of 180 kV, a current of 170 nA, an operating voltage of 120 V, and a voxel size of 2.2 μm. After the analysis formed of 1100 projections, the stack of images of 2043² pixels in size is processed to construct the volume by using Datos|X reconstruction software (Phoenix X-Ray). Segmentation and visualization were carried out by using VGSTUDIO MAX software (Volume Graphics). First, a non-linear diffusion filter was applied to reduce the noise. Second, segmentation of the volume allowed the separation by using the density difference of the silicate melt, the sulphide globule and the gas bubbles. Silicate minerals could not be differentiated from the silicate glass, due to the low density contrast between the two phases.

## Data availability
The datasets generated in this study are provided in the published article and its Supplementary Information file.

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

## Acknowledgements

This research received funding from the French agency for research (ANR project #12-JS06-0009-01), the "Equipement d'Excellence" Planex of the University of Orléans (France), and the programme TelluS of the Institut National des Sciences de l'Univers, CNRS (G.I.M.). We are grateful to Nadezhda A. Krivolutskaya, Sergey F. Sluzhenikin, and the leaders of the Noril'sk field trip of the 12th International Platinum Symposium for their precious help in sampling the starting material of this study. We thank Patricia Benoist-Julliot, Sylvain Janiec, Ida Di Carlo and Saskia Erdmann for their assistance with the SEM, EMP and LA-ICP-MS.

## Author contributions

G.I.M. and M.L.V. conceived and performed the experimental work. G.I.M. performed SEM, EMP, and LA-ICP-MS analyses. G.I.M., M.L.V., and S.J.B. wrote the manuscript. B.M.G. and L.A. performed HRXCT analyses, provided images for Fig. 3, and contributed to the writing of the Methods section. L.A. provided the final version of the figures.

## Competing interests

The authors declare no competing interests.
