## [Peer Review File · Nature Communications]

The critical role of magma degassing in sulphide melt mobility and metal enrichmentReviewers' Comments:

Reviewer #1:

Remarks to the Author:

Comments on "The critical role of magma degassing in sulphide melt mobility and metal enrichment." by Iacono-Marziano et al.

SUMMARY

In this paper, the authors report new high-temperature and high-pressure experiments designed to simulate degassing of sulphide-bearing mafic magmas and study the interactions of gas bubbles and sulphide melt droplets. High-resolution X-ray computed tomography images are presented and show the 3D spatial relationships of the compound drops. The results have potentially important implications for our understanding of magmatic sulphide ore genesis. They show that gas bubbles promote coalescence of sulphide droplets and suggest that sulphide consumption through SO₂ degassing can lead to metal enrichment in the remaining sulphide droplets. This represents a significant advance in line with recent work (e.g., by Mungall and others). The case for sulphide coalescence is mostly done qualitatively looking at images — it seems the authors should be able to exploit their μ CT data more quantitatively, by looking at number and size of bubbles and sulphide blebs as a function of the amount of fluid phase.

As far as the presentation of the manuscript goes, there is room for much improvement. It is mostly well written, but it is quite short, with only 4 display items, and it does not follow the structure of Nature Communications articles (Abstract, Introduction, Results, Discussion, and Methods). Figures need work. I suggest the authors expand and restructure their article to strengthen their case. See details below.

MAIN COMMENTS

Quantitative use of μ CT images

The authors present nice μ CT images, but use them only qualitatively, e.g., to argue for sulphide coalescence. However, with such data, it should be possible to compute the number and size of the various objects (here bubbles and sulphide blobs) to make a much stronger case. For example, if degassing indeed favours sulphide coalescence then the size of the sulphides should increase, while their number relative to the number of gas bubbles should decrease. I invite the authors to better exploit their μ CT data. Additional figures presenting these results should be added.

Manuscript structure

The paper reads quite well but would benefit from a clearer structure. Right now the section headings are:

- (1) Introduction (heading missing)
- (2) New experiments
- (3) Sulphide coalescence
- (4) Metal enrichment
- (5) Formation of platinum group minerals
- (6) Implications for ore genesis

In the current manuscript, both results and discussion of results are presented in (3) to (5), while (6) serves as discussion only. This does not follow the journal guidelines. I suggest one of the two following restructures:

- (1) Introduction
- (2) Results
 - a. Degassing experiments
 - b. High-resolution X-ray computed tomography
- (3) Discussion
 - a. Sulphide coalescence
 - b. Metal enrichment
 - c. Formation of platinum group minerals
 - d. Comparison with the Noril'sk-Talnakh deposit and implications for ore genesis

In this restructure, only observations (no interpretations, argumentations) would be presented under (2). This would involve a rearrangement of the manuscript.

Alternatively, if presenting results and discussion of results together is allowable as per the journal style, then I would suggest the following:

- (1) Introduction
- (2) Results and discussion
 - a. Degassing experiments
 - b. Sulphide coalescence
 - c. Metal enrichment
 - d. Formation of platinum group minerals
 - e. Comparison with the Noril'sk-Talnakh deposit and implications for ore genesis

In this case, the current structure is largely kept, and this would only require a slight tweaking of the manuscript structure.

In any case, I recommend the following:

-After L62, at the end of the Introduction, the authors should have a brief statement of the objectives of this paper, in the light of the scientific problem they have just introduced. This is sort of done in the first sentence of the "New experiments" section, but it could be made stronger and should appear at the end of the Introduction, not in the Results.

-In the beginning of the Results section (currently called New experiments), the reader needs to know a little more about your experiments, so I recommend including Table S1 as part of the main text. And please state from the beginning what starting material/bulk composition (Noril'sk) you have used and why. This currently comes in near the end of the paper, way too late.

-While L65-78 are essentially a brief summary of the experimental design, L79-84 is a (too) short paragraph where the authors state their main results. Please start this paragraph by "We find..." or something like this to make it clear that here begins the description of the results.

MINOR COMMENTS

L33-36. Cite the "few available quantitative studies".

L45: Define what is meant by "sulfide upgrading".

L69: "its solubility primarily"

L78: "fayalite-magnetite-quartz oxygen fugacity buffer"

L87: "samples are bubble-bearing"

L89: "analysis shows the strong"

L107: "in the magma-coal interaction experiment"

L130: Shouldn't the word "undecompressed" be replaced by "isobaric" here and throughout the text and figures? Undecompressed is a weird word, and to me sounds like something has been decompressed before and now is being recompressed.

L141: If degassing favors sulphide consumption/breakdown to the fluid phase, then how could it favor the formation of magmatic sulfide ore deposits? This is something that perhaps needs a little more explanation and justification in the text.

L170-173: Perhaps worth mentioning that it is common in natural volcanic samples that have undergone extensive degassing to find sulphides only has inclusions in the phenocrysts, and not in the groundmass/glass (e.g., Beaudry et al, 2018; Berlo et al., 2014)?

L214 "(see ref. 18 for a review)"

FIGURES

Please adapt all your figure to follow journal style. Fig. 3 and Fig. S1 appear to come straight out of MS Excel without much formatting. Please improve.

Fig. 1: A: Homogeneously distributed sulphide droplets? They seem to cluster on the margins of the sample. Please show in A the portion of the sample enlarged in B.

L394: "did not experience"

SUPPLEMENTARY INFORMATION

Analytical methods, first line: In what lab was the EPMA work done?

Table 1: Table 1 should be presented in the main text, as one of the display items.

In fO₂ column, write ΔFMQ for consistency with main text and because the redox reaction is fayalite = magnetite + quartz. What does the notation +X.X/Y.Y mean (e.g., +0.6/1.6 in first row of table)?

Does this represent the range of fO₂, e.g., from FMQ + 0.6 to +1.6, or is this the uncertainty in the calculations, e.g., FMQ + 0.6 (±1.6). Please provide clear explanation in table footnotes. In the Fluid phase column, what is the meaning of the digit in parenthesis. Please provide clear explanation in table footnotes.

Reviewer #2:

Remarks to the Author:

Dear Giada Iacono-Marziano et al.,

I have reviewed your manuscript submitted to Nature Communications entitled The critical role of magma degassing in sulphide melt mobility and metal enrichment. I found the paper to be very well written. The experimental work beautifully documents interactions between sulfide liquid and a fluid phase under magmatic conditions and the loss of S from the sulfide liquid by degassing under static experimental conditions.

To improve the manuscript. I would appreciate further discussion of how the sulfides in the low-S PGE ores in the Norilsk deposit reach such high tenors. I do not see a strong connection between high tenor PGE ores and the results of your experiments in the current form of the manuscript. How scalable are your results from very low R factors (in your experiment) to very high R factors (in the Norilsk deposits)? Is transport of compound droplets followed by degassing (to lose some sulfur) sufficient to reach the extremely high tenors found in the deposit? Is this transport happening in a mush or in crystal-free magma chamber? These processes seem ambiguous in the current version of the manuscript. A few sentences fully elucidating the connection between the exsolution of sulfide liquid, transport/equilibration of sulfide liquid with silicate melt, and degassing would be useful.

I have annotated a pdf with my additional comments, questions, and concerns.

Best regards,

Chris Jenkins

Reviewer #3:

Remarks to the Author:

Review of NCOMMS-21-35730-T

The critical role of magma degassing in sulphide melt mobility and metal enrichment

By Iacono-Marziano et al

Summary

This is a very interesting paper advancing our understanding of the role volatiles play in magmatic sulfide ore genesis – traditionally, as pointed out by the authors, an overlooked control. It builds on a number of recent empirical and experimental works that have investigated the role of volatiles as ‘compound droplets’ with sulfide melt droplets. In those studies, the focus has been on the physical transport implications. Where this paper really takes us forward is that it also shows that not only do the bubbles play a role (or not!) in favourable transport... they can also dissolve S, and thus upgrade the metal tenor of the sulfide droplets. For this reason, this paper is very significant and I would like to see it published in Nature Communications. I have some suggestions/comments that the authors should consider to improve the paper, and perhaps the scope of the results.

Major comments

- The focus here is on upper crustal conditions, and bubbles that are largely hydrous (as per most experimental work – e.g. Yao and Mungall studies). It would expand the scope of the implications if the authors were at least to comment on whether this process can happen in the lower crust. Yao and Mungall (EPSL 2020) state that the dynamics of compound droplet formation are non pressure sensitive, and Blanks et al (NComms 2020) propose compound droplets with supercritical CO₂ as the bubble phase. Whilst the upward migration possibilities have been established, what do the authors new findings on metal tenors mean for compound droplets that are largely CO₂? I suspect S would be less soluble, but I may be wrong... would the upgrading by CO₂ bubbles be less or more prevalent than that shown? I am not suggesting the authors do any new experiments, but just add some comment as to what the implications may be for deeper, more CO₂ rich compound droplets.
- Following on from this, whilst the relevance to Norilsk is very strong, it would be good to hear about other examples where this may well have happened, otherwise it could be seen as being specific to Norilsk. The implications of this being potentially widespread makes this contribution stronger so taking the reader out from the case study of Norilsk into some other areas would do that. Line 248 for example – expand on the references that at the moment are just simply cited without comment. Similarly.. what about mantle derived volatiles? I speak here primarily of the CO₂ referred to above... is the subject of the paper a crustal contamination feature, or ANY volatile phases?
- What is the Pt-rich phase? I know this was done by EPMA but what other elements are there? Any other PGE? One might expect some Pd too if these are from Norilsk. Is the Pt phase an alloy, telluride, arsenide? It would be useful to know that – especially as the authors do point out that they have used Pt crucibles (though I take their point only one sample shows Pt nuggets). If it is just Pt... then that is a little questionable. If it is a common PGM phase, then that would take away some uncertainty. If the authors could check (which should only take an hour on an SEM or suchlike, that would be a useful addition.
- Figure 4 should have an image of the massive sulfides as well for completeness (with line 220-223).
- There is inconsistent use of chemical symbol and full spelling for sulphur. To be consistent with other elements referred to, use S throughout.

Minor comments

Line Comment

12 Personal preference here, but to avoid the use of "we" in this case (I know it means society, but could mean the authors themselves!), rephrase to "A major high quality deposit of these metals needs to be discovered every year..."

34 Add references to the statement "few available quantitative studies"

52-54 I'm not sure of the significance of this sentence. It doesn't seem to link obviously to the arguments. Perhaps a clarify statement with context to the problem being investigated would help

62 To avoid a short statement at the end, "...magma; a property well known in metallurgy"

79-84 This paragraph seems out of place and introduces some data without reference to figures – including the PGM. I don't actually think this is necessary and can probably be deleted, or moved to after the results.

87 Change 'and' to 'are'

213-219 Not clear if this paragraph is referring to Norilsk or more general deposits.

241 Where do the bubbles go if they detach from the sulfide

Fig. 1 Label the Pt phase on G

David Holwell

University of Leicester, UK

07/10/2021

We sincerely appreciated the constructive feedback given by the three reviewers that helped us to significantly improve the manuscript. We incorporated all their comments/suggestions, as shown in detail below. Hereafter a brief summary of the main changes:

- Following the suggestions of Reviewer #1, we modified the manuscript structure and we presented more quantitative results from microCT analyses (Fig.4 and Supplementary Table 4).
- As proposed by Reviewer #2, we introduced a section to further discuss how Noril'sk ores may have reached their high tenors (lines 296-326, Fig.7 and Supplementary Fig.2).
- In agreement with the suggestions of Reviewer # 3, we described other contexts where the processes we deal with may have occurred (lines 346-354 and 365-371), and present possible implications for the lower crust (lines 378-388).

Reviewer comments

Reviewer #1 (Remarks to the Author):

Comments on "The critical role of magma degassing in sulphide melt mobility and metal enrichment." by Iacono-Marziano et al.

SUMMARY

In this paper, the authors report new high-temperature and high-pressure experiments designed to simulate degassing of sulphide-bearing mafic magmas and study the interactions of gas bubbles and sulphide melt droplets. High-resolution X-ray computed tomography images are presented and show the 3D spatial relationships of the compound drops. The results have potentially important implications for our understanding of magmatic sulphide ore genesis. They show that gas bubbles promote coalescence of sulphide droplets and suggest that sulphide consumption through SO₂ degassing can lead to metal enrichment in the remaining sulphide droplets. This represents a significant advance in line with recent work (e.g., by Mungall and others). The case for sulphide coalescence is mostly done qualitatively looking at images — it seems the authors should be able to exploit their μ CT data more quantitatively, by looking at number and size of bubbles and sulphide blebs as a function of the amount of fluid phase.

As far as the presentation of the manuscript goes, there is room for much improvement. It is mostly well written, but it is quite short, with only 4 display items, and it does not follow the structure of Nature Communications articles (Abstract, Introduction, Results, Discussion, and Methods). Figures need work. I suggest the authors expand and restructure their article to strengthen their case. See details below.

MAIN COMMENTS

Quantitative use of μ CT images

The authors present nice μ CT images, but use them only qualitatively, e.g., to argue for sulphide coalescence. **MicroCT statistics are already used quantitatively to confirm the estimation of the proportion of sulfide melt by mass balance (Table 1).**

However, with such data, it should be possible to compute the number and size of the various objects (here bubbles and sulphide blobs) to make a much stronger case. For example, if degassing indeed favours sulphide coalescence then the size of the sulphides should increase, while their number relative to the number of gas bubbles should decrease. I invite the authors to better exploit their μ CT data. Additional figures presenting these results should be added. **Two new diagrams (Fig. 4) are now presented and discussed in the text (lines 128-130 and 149-151). They show that the sulphide blebs within decompressed samples attain larger volumes and are less numerous than those in isobaric samples. MicroCT statistics are also presented in Supplementary Table 4.**

Manuscript structure

The paper reads quite well but would benefit from a clearer structure. Right now the section headings are:

- (1) Introduction (heading missing)
- (2) New experiments
- (3) Sulphide coalescence
- (4) Metal enrichment
- (5) Formation of platinum group minerals
- (6) Implications for ore genesis

In the current manuscript, both results and discussion of results are presented in (3) to (5), while (6) serves as discussion only. This does not follow the journal guidelines. I suggest one of the two following restructures:

- (1) Introduction
- (2) Results
 - a. Degassing experiments
 - b. High-resolution X-ray computed tomography
- (3) Discussion
 - a. Sulphide coalescence
 - b. Metal enrichment
 - c. Formation of platinum group minerals
 - d. Comparison with the Noril'sk-Talnakh deposit and implications for ore genesis

In this restructure, only observations (no interpretations, argumentations) would be presented under (2). This would involve a rearrangement of the manuscript. **The manuscript has been rearranged following this restructure.**

Alternatively, if presenting results and discussion of results together is allowable as per the journal style, then I would suggest the following:

- (1) Introduction
- (2) Results and discussion
 - a. Degassing experiments
 - b. Sulphide coalescence
 - c. Metal enrichment
 - d. Formation of platinum group minerals

e. Comparison with the Noril'sk-Talnakh deposit and implications for ore genesis

In this case, the current structure is largely kept, and this would only require a slight tweaking of the manuscript structure.

In any case, I recommend the following:

-After L62, at the end of the Introduction, the authors should have a brief statement of the objectives of this paper, in the light of the scientific problem they have just introduced. This is sort of done in the first sentence of the "New experiments" section, but it could be made stronger and should appear at the end of the Introduction, not in the Results. **Added (lines 67-72)**

-In the beginning of the Results section (currently called New experiments), the reader needs to know a little more about your experiments, so I recommend including Table S1 as part of the main text. **Changed**

And please state from the beginning what starting material/bulk composition (Noril'sk) you have used and why. This currently comes in near the end of the paper, way too late. **Added (lines 76-78)**

-While L65-78 are essentially a brief summary of the experimental design, L79-84 is a (too) short paragraph where the authors state their main results. Please start this paragraph by "We find..." or something like this to make it clear that here begins the description of the results. **The paragraph has been reorganized and enriched (lines 83-114).**

MINOR COMMENTS

L33-36. Cite the "few available quantitative studies". **Added**

L45: Define what is meant by "sulfide upgrading". **Added**

L69: "its solubility primarily" **Modified**

L78: "fayalite-magnetite-quartz oxygen fugacity buffer" **Added**

L87: "samples are bubble-bearing" **Modified**

L89: "analysis shows the strong" **Modified**

L107: "in the magma-coal interaction experiment" **Added**

L130: Shouldn't the word "undecompressed" be replaced by "isobaric" here and throughout the text and figures? Undecompressed is a weird word, and to me sounds like something has been decompressed before and now is being recompressed. **Modified**

L141: If degassing favors sulphide consumption/breakdown to the fluid phase, then how could it **favor the formation of magmatic sulfide ore deposits?** This is something that perhaps needs a little more explanation and justification in the text. **Clarified at lines 257-264 and in Figure 6. The proportion of fluid phase controls whether coalescence and accumulation dominate (low fluid proportions), or consumption and metal enrichment dominate (high fluid proportions). This leads to**

the formation of different types of ores (i.e., massive and metal poorer or disseminated and metal richer)

L170-173: Perhaps worth mentioning that it is common in natural volcanic samples that have undergone extensive degassing to find sulphides only has inclusions in the phenocrysts, and not in the groundmass/glass (e.g., Beaudry et al, 2018; Berlo et al., 2014)? **Added**

L214 "(see ref. 18 for a review)" **Modified**

FIGURES

Please adapt all your figure to follow journal style. Fig. 3 and Fig. S1 appear to come straight out of MS Excel without much formatting. Please improve. **Improved**

Fig. 1: A: Homogeneously distributed sulphide droplets? They seem to cluster on the margins of the sample. **Removed** Please show in A the portion of the sample enlarged in B. **Added**

L394: "did not experience" **Modified**

SUPPLEMENTARY INFORMATION

Analytical methods, first line: In what lab was the EPMA work done? **Added, Institut des Sciences de la Terre d'Orléans.**

Table 1: Table 1 should be presented in the main text, as one of the display items. **Added**

In fO₂ column, write Δ FMQ for consistency with main text and because the redox reaction is fayalite = magnetite + quartz. **Modified** What does the notation +X.X/Y.Y mean (e.g., +0.6/1.6 in first row of table)? Does this represent the range of fO₂, e.g., from FMQ + 0.6 to +1.6, or is this the uncertainty in the calculations, e.g., FMQ + 0.6 (\pm 1.6). Please provide clear explanation in table footnotes. In the Fluid phase column, what is the meaning of the digit in parenthesis. Please provide clear explanation in table footnotes. **Clarified**

Reviewer #2 (Remarks to the Author):

Dear Giada Iacono-Marziano et al.,

I have reviewed your manuscript submitted to Nature Communications entitled The critical role of magma degassing in sulphide melt mobility and metal enrichment. I found the paper to be very well written. The experimental work beautifully documents interactions between sulfide liquid and a fluid phase under magmatic conditions and the loss of S from the sulfide liquid by degassing under static experimental conditions.

To improve the manuscript. I would appreciate further discussion of how the sulfides in the low-S PGE ores in the Norilsk deposit reach such high tenors. I do not see a strong connection between

high tenor PGE ores and the results of your experiments in the current form of the manuscript. How scalable are your results from very low R factors (in your experiment) to very high R factors (in the Noril'sk deposits)? Is transport of compound droplets followed by degassing (to lose some sulfur) sufficient to reach the extremely high tenors found in the deposit? Is this transport happening in a mush or in crystal-free magma chamber? These processes seem ambiguous in the current version of the manuscript. A few sentences fully elucidating the connection between the exsolution of sulfide liquid, transport/equilibration of sulfide liquid with silicate melt, and degassing would be useful.

A paragraph (lines 296-326) and two figures (Fig. 7, and Supplementary Fig.2) were introduced to clarify these points. R-factor calculations are shown, together with Ni-Cu and Pt-Pd contents of Noril'sk massive and disseminated sulphides. Extreme sulphide consumption (69% on average and up to 87%), may explain Ni and Cu contents and most of Pt and Pd contents of globular ores. A contribution from sulphide-silicate interaction is also expected, due to the upward transport of compound drops that bring them in contact with undepleted magma.

Unfortunately, no precise data are available for low PGE ores, but rough estimations indicate higher Pt and Pd contents, which strongly suggest the formation of PGM by sulphide loss as a critical process. PGE formation cannot be simulated by R-factor calculations because doesn't imply a sulphide melt anymore.

I have annotated a pdf with my additional comments, questions, and concerns.

Comments copied from the annotated pdf:

Lines 30-33: Long sentence, perhaps split into two. Do they need efficient mechanisms for sulfide segregation or sulfide accumulation? Or both? **Modified**

Lines 148-149: This isn't really the same though is it? In the sulfide upgrading stories, we're dissolving some resident sulfide liquid back into the silicate melt AND adding HSE to the remaining sulfide liquid. That addition process is important to get to the extremely high tenor sulfide liquid. In your model, sulfide is dissolving which gives your sulfide the appearance of an upgraded tenor. Not really sulfide upgrading as much as sulfide loss. **Clarified (lines 206-209)**

Lines 151-154: You only show one model curve for each diagram but report a range of values. Can you specify what Kds you used for each element? **The full range of values is now reported in the text and curves calculated with the minimum and maximum value are shown in the figure (differences are negligible).**

Lines 154-156: I understand that the S leaving the system by partitioning into the fluid phase is going to increase the apparent R factor of your sulfide liquid, but I don't understand how this is going to form a high-tenor deposit like the low-S PGE deposits you describe. For the low-S PGE deposits Sluzhenikin et al (2020) Econ. Geol. Report grades of 10-50g/t PGE with tenors up to 1400-2500 g/t total PGE in 100% sulfide. You can get to those tenors—or higher at the point the system is just above SCSS—by removing S until just before the silicate melt becomes S undersaturated. I suspect that these sulfides already needed to reach very high R factors by more conventional means before they were transported on compound droplets (I believe this is what Yao et al., 2019 JPet also proposed). Your model leaves me with the impression that assimilation of S-bearing country rocks is necessary to reach SCSS & to form the droplets and the vapor bubbles more or less at the level of the magma chamber and at the same time. It seems hard to understand how those sulfide droplets could

equilibrate with enough PGE-bearing silicate melt to form the deposits in that scenario. You don't address this mass balance issue at any point in the manuscript. It would be nice to know how you think your sulfides are interacting with enough silicate melt to reach very high tenors before some of the S is removed during degassing. If they're just floating upwards in a magma chamber is that sufficient reach the high grades and tenors observed in Noril'sk? I completely buy into the idea that compound droplets are moving sulfide liquid around, but it seems that the sulfide liquid should also have high R factors before S is removed. A sentence or two explaining how this all works would be nice. **As explained above, a paragraph (lines 296-326) and figure 7 were introduced to clarify these points. The details of calculations for Pt and Pd are presented in the Supplementary Information, as we used data from the literature and not from our experiments.**

Lines 238-241: Is this process all happening in one large magma chamber? **Nori'sk-Talnakh intrusions are not large magma chambers (clarified at line 266).** Heavy compound droplets are sinking to the bottom whereas the disseminated and low-S ores are compound droplets that floated upwards? **Figure 6 has been modified to clarify how ore distribution is related to the described processes.** Are remnants of bubbles found in the massive ores? **Clarified**

Fig. 1A: Why are the sulfide droplets isolated to the sides of the capsule? **Explained (lines 85-87).**

Fig.4: Can element color wheels be added to this figure to show what element each color is? E.g., red Cr, blue S, green ?? in top map. **The figure has been modified**

Best regards,

Chris Jenkins

Reviewer #3 (Remarks to the Author):

Review of NCOMMS-21-35730-T

The critical role of magma degassing in sulphide melt mobility and metal enrichment

By Iacono-Marziano et al

Summary

This is a very interesting paper advancing our understanding of the role volatiles play in magmatic sulfide ore genesis – traditionally, as pointed out by the authors, an overlooked control. It builds on a number of recent empirical and experimental works that have investigated the role of volatiles as 'compound droplets' with sulfide melt droplets. In those studies, the focus has been on the physical transport implications. Where this paper really takes us forward is that it also shows that not only do the bubbles play a role (or not!) in favourable transport... they can also dissolve S, and thus upgrade the metal tenor of the sulfide droplets. For this reason, this paper is very significant and I would like to see it published in Nature Communications. I have some suggestions/comments that the authors should consider to improve the paper, and perhaps the scope of the results.

Major comments

- The focus here is on upper crustal conditions, and bubbles that are largely hydrous (as per most experimental work – e.g. Yao and Mungall studies). It would expand the scope of the implications if the authors were at least to comment on whether this process can happen in the lower crust. Yao

and Mungall (EPSL 2020) state that the dynamics of compound droplet formation are non pressure sensitive, and Blanks et al (NComms 2020) propose compound droplets with supercritical CO₂ as the bubble phase. Whilst the upward migration possibilities have been established, what do the authors new findings on metal tenors mean for compound droplets that are largely CO₂? I suspect S would be less soluble, but I may be wrong... would the upgrading by CO₂ bubbles be less or more prevalent than that shown? I am not suggesting the authors do any new experiments, but just add some comment as to what the implications may be for deeper, more CO₂ rich compound droplets.

Discussed at lines 378-388. Our data suggest that the amount of fluid phase is the first order parameter in determining S degassing and therefore sulphide consumption. If this is confirmed for CO₂-dominated systems and higher pressures (further experiments are needed), CO₂ degassing is likely to have a weaker effect on S degassing, as CO₂ solubility, even at lower crustal conditions, is generally lower than H₂O solubility.

- Following on from this, whilst the relevance to Norilsk is very strong, it would be good to hear about other examples where this may well have happened, otherwise it could be seen as being specific to Norilsk. The implications of this being potentially widespread makes this contribution stronger so taking the reader out from the case study of Norilsk into some other areas would do that. Line 248 for example – expand on the references that at the moment are just simply cited without comment. Similarly.. what about mantle derived volatiles? I speak here primarily of the CO₂ referred to above... is the subject of the paper a crustal contamination feature, or ANY volatile phases?

A description of other examples has been added (lines 346-354 and 365-371), and the mantle/crustal derivation of volatile better discussed.

- What is the Pt-rich phase? I know this was done by EPMA but what other elements are there? Any other PGE? One might expect some Pd too if these are from Norilsk. Is the Pt phase an alloy, telluride, arsenide? It would be useful to know that – especially as the authors do point out that they have used Pt crucibles (though I take their point only one sample shows Pt nuggets). If it is just Pt... then that is a little questionable. If it is a common PGM phase, then that would take away some uncertainty. If the authors could check (which should only take an hour on an SEM or suchlike, that would be a useful addition. Some information has been added (lines 92-95), but the extremely small size of the particles (<1 μm) does not allow a full characterization.

- Figure 4 should have an image of the massive sulfides as well for completeness (with line 220-223. Added

- There is inconsistent use of chemical symbol and full spelling for sulphur. To be consistent with other elements referred to, use S throughout. Modified

Minor comments

Line Comment

12 Personal preference here, but to avoid the use of “we” in this case (I know it means society, but could mean the authors themselves!), rephrase to “A major high quality deposit of these metals needs to be discovered every year...” Modified

34 Add references to the statement “few available quantitative studies” **Added**

52-54 I’m not sure of the significance of this sentence. It doesn’t seem to link obviously to the arguments. Perhaps a clarify statement with context to the problem being investigated would help **Clarified**

62 To avoid a short statement at the end, “...magma; a property well known in metallurgy” **Modified**

79-84 This paragraph seems out of place and introduces some data without refeence to figures – including the PGM. I don’t actually think this is necessary and can probably be deleted, or moved to after the results. **Removed**

87 Change ‘and’ to ‘are’ **Modified**

213-219 Not clear if this paragraph is referring to Norilsk or more general deposits. **Clarified**

241 Where do the bubbles go if they detach from the sulfide

Added at lines 293-294. They escaped into the host rocks contributing to the formation of the metasomatic aureoles.

Fig. 1 Label the Pt phase on G **Added**

David Holwell
University of Leicester, UK
07/10/2021

Reviewers' Comments:

Reviewer #1:

Remarks to the Author:

SUMMARY

The authors have addressed most of the points previously raised, although their figures 2, 4, 5 should still be improved significantly to bring them up to journal standards. Also, wouldn't scatter plots be more useful than bar charts for Fig. 4?

MINOR COMMENTS

L12: "sub-volcanic magma plumbing systems."

L21: "process producing the fluid phase."

L22-23: "...processes demonstrated experimentally."

L47: "...being upgraded — such that their concentrations in metals have been increased — by scavenging..."

L76-78: Need to call table listing the starting composition and cite references that documented the olivine gabbro-dolerite of the Noril'sk 1 intrusion.

L79: "isobaric or decompressed (with and without decompression)." This is confusing. Please rewrite.

L87: "(generally lower than 10%)" What does this refer to? Fe-loss? Unclear, please rephrase.

L87-90: Suggest rephrase/reorder as: "Sulphide droplets of the decompressed samples are generally larger than those of the isobaric ones (Fig. 1a,c). In decompressed samples, sulphide droplets located in the sample's upper part are larger than those occurring the lower part of the capsule (Fig. 1c)."

L105: "with decompressed samples generally..."

L102-108: If I understand this part correctly, it relies on HRXCT data to calculate the proportion of sulphide melt. If so, shouldn't this be presented in the following section, when presenting the XRXCT results?

L127-129: This seems redundant.

L176: "which presents large sulphide blebs"

L254-326: This is a single paragraph spanning 2+ pages. Suggest break down in multiple paragraphs at the appropriate places.

L296: For consistency in spelling, write "sulphides".

L313: "under more oxidising"

L342: Delete both commas in this line.

Reviewer #2:

Remarks to the Author:

Dear Iacono-Marziano and coauthors,

I have reviewed your revised manuscript and feel that you've responded in full to my original comments and concerns. I believe your manuscript should be accepted to Nature Communications after some very minor editorial changes to some figures and the text. I attach annotated pdfs that include the typos that I caught during the review.

Best regards,

Chris

Reviewer #3:

Remarks to the Author:

I am happy that the authors have addressed all of my initial comments/suggestions and have a very good paper that I would like to see published.

We thank the reviewers for their additional suggestions that further improve the quality of the paper. We incorporated most of the comments/suggestions, as shown in detail below. We confirm that we saw the mark ups from referee #2 in the attached PDFs, which are copied below. We also revised our figures to make them somewhat more appealing and to comply with the journal policies.

Reviewer #1 (Remarks to the Author):

SUMMARY

The authors have addressed most of the points previously raised, although their figures 2, 4, 5 should still be improved significantly to bring them up to journal standards. Also, wouldn't scatter plots be more useful than bar charts for Fig. 4? **Figures 2, 4, 5, 6 and 7 were improved. Sorry but we cannot see how to transform the bar charts into scatter plots, without losing part of the useful information.**

MINOR COMMENTS

L12: "sub-volcanic magma plumbing systems." **Changed**

L21: "process producing the fluid phase." **Changed**

L22-23: "...processes demonstrated experimentally." **Changed**

L47: "...being upgraded — such that their concentrations in metals have been increased — by scavenging..." **Changed**

L76-78: Need to call table listing the starting composition and cite references that documented the olivine gabbro-dolerite of the Noril'sk 1 intrusion. **Added**

L79: "isobaric or decompressed (with and without decompression)." This is confusing. Please rewrite. **With and without decompression was deleted**

L87: "(generally lower than 10%)" What does this refer to? Fe-loss? Unclear, please rephrase. **Changed**

L87-90: Suggest rephrase/reorder as: "Sulphide droplets of the decompressed samples are generally larger than those of the isobaric ones (Fig. 1a,c). In decompressed samples, sulphide droplets located in the sample's upper part are larger than those occurring the lower part of the capsule (Fig. 1c)." **Changed**

L105: "with decompressed samples generally..." **Changed**

L102-108: If I understand this part correctly, it relies on HRXCT data to calculate the proportion of sulphide melt. If so, shouldn't this be presented in the following section, when presenting the XRXCT results? **This part does not rely on HRXCT data, the sentence has been modified into "We also used an additional sample from a previous experiment, in which the fluid phase was generated by the interaction with coal."**

L127-129: This seems redundant. **Modified into "HRXCT renderings show that the sulphide droplets connected to the same fluid bubble may coalesce, forming larger blebs (Fig. 3c)."**

L176: "which presents large sulphide blebs" **Changed**

L254-326: This is a single paragraph spanning 2+ pages. Suggest break down in multiple paragraphs at the appropriate places. **No subheadings are allowed in the Discussion section. All subheadings have been removed.**

L296: For consistency in spelling, write "sulphides". **Changed**

L313: "under more oxidising" **Changed**

L342: Delete both commas in this line. **Changed**

Reviewer #2 (Remarks to the Author):

Dear Iacono-Marziano and coauthors,

I have reviewed your revised manuscript and feel that you've responded in full to my original comments and concerns. I believe your manuscript should be accepted to Nature Communications after some very minor editorial changes to some figures and the text. I attach annotated pdfs that include the typos that I caught during the review.

Best regards,

Chris

Line 306: ,R or (i.e. R-factor) **Changed**

Line 317: more oxidizing than **Corrected**

Fig. 4: I assume that "isobaric" indicates that MLV16 & 17 were the isobaric samples. The label could be reformatted to better indicate this. Should this be number (no.)? What are the units on this? μm^{-2} ? **Modified**

Fig.5: I can't tell what the solid versus dashed lines refer to, I suspect different D values for sulf/sil liquids **Indeed, this has been clarified in the figure legend**

Fig.7: initial Ni and Cu concentration in the parental magma of 120 ppm and 150 ppm, respectively **Changed**

Reviewer #3 (Remarks to the Author):

I am happy that the authors have addressed all of my initial comments/suggestions and have a very good paper that I would like to see published.